# LEARNING TEXT-DRIVEN 3D HUMAN MOTION GENERATION FROM 3D-FREE WEB VIDEOS

## ABSTRACT

Text-driven 3D human motion generation has gained attention for synthesizing complex movements from textual descriptions. Traditional approaches depend on expensive 3D motion capture, which restricts motion diversity, whereas 2D human videos provide abundant and accessible data. However, the absence of large-scale annotated 2D motion datasets and the challenge of generating 3D motion from 2D data remain unresolved. To address this, we introduce **MotionWeb**, a dataset comprising over 100k motion clips, 17 million frames, and 160 hours of data, with 2D keypoints extracted using state-of-the-art pose estimation models, significantly reducing annotation costs. We further propose Keypoint To Motion (**K2M**), an efficient framework for text-driven 3D motion generation leveraging 2D supervision without requiring 3D annotations. Experiments show that our method efficiently generates realistic 3D motion with improved both quality and diversity using large-scale 2D supervision.

## 1 INTRODUCTION

Text-driven 3D human motion generation bridges natural language understanding and motion synthesis (Zhu et al., 2023b), enabling realistic motion generation from text. This technology enhances virtual reality, gaming, and animation by reducing manual effort and personalizing experiences.

However, data scarcity remains a persistent challenge in text-driven 3D human motion generation. Collecting 3D motion data typically requires controlled lab environments, limiting motion diversity, while outdoor data lack accurate 3D annotations. Moreover, constructing new text-annotated 3D datasets is costly due to expensive MoCap equipment and complex post-processing (Mahmood et al., 2019). This raises the question of how to leverage web-scale data for training. Compared to 3D motion sequences, 2D video data is more accessible, cost-effective, and abundant. Advances in pose estimation models have improved their accuracy, making them viable substitutes for ground-truth annotations. However, generating 3D motion directly from 2D data remains challenging due to the absence of 3D supervision, preventing loss-based optimization. Furthermore, 2D keypoints lack depth information and do not enforce biomechanical plausibility.

We constructed a large-scale dataset for 3D motion generation by aggregating over 100k high-quality 2D motion clips, totaling more than 17 million frames and 160 hours, sourced from web videos and existing datasets (Taheri et al., 2020; Lin et al., 2023). Using ViTPose (Xu et al., 2022), we extracted 2D keypoints and generated textual descriptions with AuroraCap (Chai et al., 2024). Compared to existing 3D motion datasets, **MotionWeb** is significantly larger and encompasses a broader range of outdoor activities.

Based on the **MotionWeb** dataset, we propose a novel method Keypoint To Motion (**K2M**), which only uses 2D keypoints in training while enabling text-driven 3D human motion generation. **K2M** includes two main parts: 2D-to-3D motion tokenization and Text-to-Motion transformer. For 2D-to-3D motion tokenization part, we propose MotionVQ-Adapter, an extension of residual vector quantization variational autoencoder (VQ-VAE) (Zeghidour et al., 2021; Guo et al., 2024) that integrates an adapter module. MotionVQ-Adapter first encodes the input 2D keypoint sequence into a hierarchical set of motion tokens using residual vector quantization and directly generates SMPL parameters (Loper et al., 2015), eliminating unnecessary conversions and improving efficiency. Since **K2M** requires a higher-dimensional output space, the adapter module facilitates the effective transformation and refinement of 2D motion representations. To provide supervision, we incorporate a 3D-to-2D

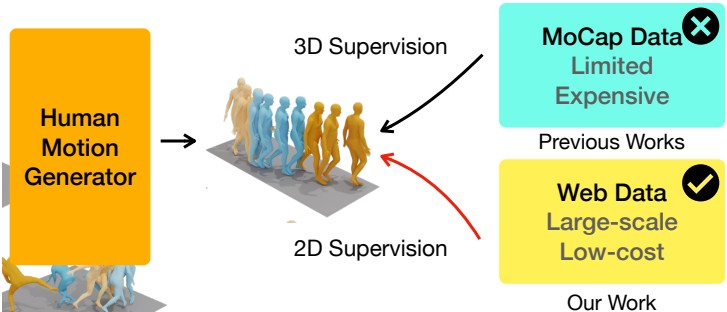

Figure 1: **Motivation.** Due to the scarcity and expense of 3D MoCap data, training a model that excels in both quality and diversity is extremely challenging. In this work, we explore how to utilize large-scale web data to achieve better performance with only 2D pseudo-label supervision.

projection model that maps the generated SMPL motion sequences back to 2D keypoints for comparison with the input. Text-to-Motion transformer generates motion tokens conditioned on textual input, using a masked transformer (Chang et al., 2022; 2023; Li et al., 2023) to predict base motion tokens and a residual transformer (Guo et al., 2024) to iteratively refine motion details through residual reconstruction. At inference time, **K2M** could directly generate SMPL parameters from textual descriptions, enabling seamless rendering without additional format conversions.

Our main contributions can be summarized as follows: (1) We introduce the **MotionWeb** dataset, which contains 101,208 motion sequences paired with 154,680 textual captions, to facilitate advances in text-driven motion generation. (2) We propose a novel method **K2M** for generating realistic 3D motions using only 2D keypoints as supervision, leveraging the MotionVQ-Adapter for efficient motion tokenization. (3) **K2M** shows strong performance on the **MotionWeb** dataset, generating realistic and natural motions with superior results across multiple evaluation metrics compared to other methods. This demonstrates that large-scale 2D pseudo-labels can effectively facilitate high-quality 3D motion generation while significantly reducing annotation costs.

## 2 RELATED WORK

### 2.1 TEXT-DRIVEN HUMAN MOTION GENERATION

Text-driven 3D human motion generation has garnered increasing research attention, with the goal of synthesizing motion sequences from natural language descriptions (Ahuja & Morency, 2019; Tevet et al., 2022a). Early approaches (Lin et al., 2018; Ahuja & Morency, 2019) focused on deterministic models. Then GAN-based methods (Wang et al., 2020) have been employed for action-conditioned motion generation. Recent advances in motion generation leverage diffusion models (Tevet et al., 2022b; Dabral et al., 2023; Chen et al., 2023; Zhang et al., 2023b; Meng et al., 2024). Approaches like MDM (Tevet et al., 2022b) employ scheduled diffusion (Ho et al., 2020) for progressive motion refinement, yielding high-quality outputs. VQ-based methods (Guo et al., 2022b; Kong et al., 2023; Zhang et al., 2023a; Zhong et al., 2023; Guo et al., 2024; Pinyoanuntapong et al., 2024; Yuan et al., 2025) discretize motion into tokens, enhancing realism. LLM integration (Brown et al., 2020; Kojima et al., 2022; Touvron et al., 2023) in text-to-motion pipelines (Radford et al., 2021; Jiang et al., 2023; Zhou et al., 2024; Zhang et al., 2024b) further improves generalization across motion types.

### 2.2 HUMAN MOTION DATASETS

Benchmark datasets for 3D human motion generation have been developed to facilitate the synthesis of realistic and diverse movements from textual descriptions. Early datasets (Troje, 2002; Mandery et al., 2015; Sigal et al., 2010; Müller et al., 2007) were fragmented and lacked standardized annotations. The KIT Motion-Language dataset (Plappert et al., 2016) pioneered multi-modal motion generation (Petrovich et al., 2022; Radford et al., 2021), while AMASS (Mahmood et al., 2019) unified 15 MoCap datasets (Sigal et al., 2010) using DMPL (Loper et al., 2015) parameterization. BABEL (Punnakkal et al., 2021), HumanML3D (Guo et al., 2022a) and SnapMoGen(Hwang

et al.) improved usability with structured textual annotations, offering sequence-level descriptions and multiple captions respectively.

However, these 3D MoCap captured datasets suffer from limited motion diversity and coarse textual descriptions. Recent efforts, such as Motion-X (Zhang et al., 2023a), MotionBase (Wang et al., 2024b), ScaMo (Lu et al., 2024), and Humanoid-X (Mao et al., 2024), have attempted to address these limitations by integrating existing 3D datasets and collecting in-the-wild 2D data to extract 3D information. While these approaches expand data scale, they often suffer from lower accuracy and less smooth transitions due to 3D-from-2D prediction challenges. Additionally, many datasets remain unreleased, limiting accessibility. These issues underscore the need for a scalable, annotation-efficient motion generation approach that leverages 2D data.

### 2.3 2D-TO-3D SUPERVISION

Recent advancements in 2D-to-3D supervision have achieved success across domains. DreamFusion (Poole et al., 2022) uses a pre-trained 2D text-to-image diffusion model to generate 3D models from text, bypassing 3D training data. Similarly, 3D-Fauna (Li et al., 2024) learns a pan-category 3D animal model from 2D images, generalizing to more than 100 species. In 3D human pose estimation, CanonPose (Wandt et al., 2021) leverages multiview consistency to estimate poses without 3D annotations, demonstrating the potential of 2D supervision for 3D tasks.

However, 2D-to-3D supervision remains challenging in human motion generation. MAS (Zhang et al., 2023a) uses a 2D motion diffusion model with a consistency block to enforce spatial coherence, while Motion-2-to-3 (Pi et al., 2024) fine-tunes a 2D motion generator with 3D data for multiview consistency. Nevertheless, MAS lacks text control, and Motion-2-to-3's reliance on 3D supervision limits its applicability to fully unsupervised settings. These limitations underscore the need for further advancements.

## 3 METHOD

### 3.1 OVERVIEW

Our goal is to generate a 3D SMPL (Loper et al., 2015) motion sequence $\mathcal{M}_{\text{SMPL}} \in \mathbb{R}^{F \times 85}$ based on the text prompt $\mathbf{T}$. The sequence consists of shape parameters $\boldsymbol{\beta} \in \mathbb{R}^{F \times 10}$, pose parameters $\boldsymbol{\theta} \in \mathbb{R}^{F \times 72}$, and global translation $\mathbf{t} \in \mathbb{R}^{F \times 3}$, where $F$ denotes the number of frames.

As shown in Figure 2, **K2M** framework consists of two main parts: 2D-to-3D Motion Tokenization and Text-to-Motion Transformer. 2D-to-3D Motion Tokenization (Section 3.2) converts 2D keypoints into discrete multi-layer motion tokens, generating 3D motion sequence $\mathcal{M}_{\text{SMPL}}$. Using a 3D-to-2D projection model, we enable 3D-free supervision (Section 3.3) through geometric losses and parameter constraints. Next, the Text-to-Motion Transformer (Section 3.4) progressively predicts motion tokens from text, which are decoded into the final motion sequence.

### 3.2 2D-TO-3D MOTION TOKENIZATION

To process the input 2D keypoint sequence $\mathcal{M}_{\text{2D}} \in \mathbb{R}^{F \times D}$, where $D$ represents the motion dimension, we propose the MotionVQ-Adapter. This module extends the residual VQ-VAE (Guo et al., 2024; Zeghidour et al., 2021) by integrating an adapter module, which enhances token representation learning while maintaining the hierarchical quantization structure for motion discretization. The decoder aggregates quantized representations across multiple layers to produce $\mathcal{M}_{\text{SMPL}}$. Furthermore, we incorporate a 3D-to-2D projection model to transform $\mathcal{M}_{\text{SMPL}}$ into 2D keypoints.

**MotionVQ-Adapter.** Unlike traditional 3D human motion generation methods, which maintain consistent input and output dimensions, our approach differs by transforming $\mathcal{M}_{\text{2D}}$ into 85-dimensional $\mathcal{M}_{\text{SMPL}}$, addressing the dimensional mismatch between input and output. Therefore, we propose adding an adapter to preprocess the input 2D data, as illustrated in Figure 2(a). The structure of our adapter draws inspiration from the Dual-Stream Spatio-Temporal Transformer (DSTformer) (Zhu et al., 2023a), which is designed to capture both spatial dependencies and the temporal

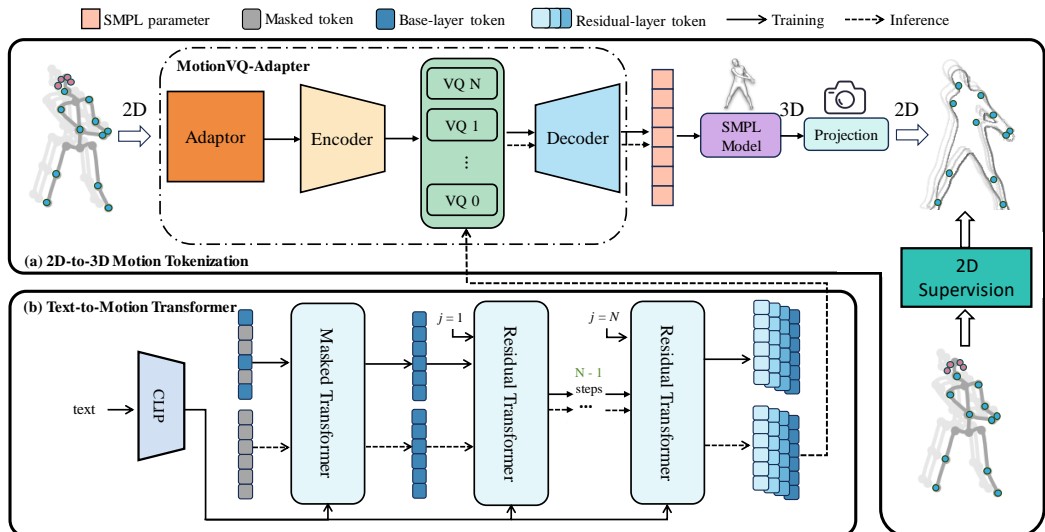

Figure 2: **Architecture of K2M.** Our model comprises: (a) 2D-to-3D Tokenization: MotionVQ-Adapter integrates an adapter into a residual VQ-VAE, converting 2D keypoints into hierarchical motion tokens and 3D SMPL representations, refined via 3D-to-2D projection. (b) Text-to-Motion Generation: A Masked Transformer predicts base-layer tokens from text, while a Residual Transformer predicts residual-layer tokens. (c) Inference: From masked base tokens and text embeddings, the transformer iteratively predicts all tokens, outputting SMPL sequences via the codebook decoder.

dynamics of pose sequences, leading to a richer representation, which is crucial for our motion generation framework.

During training, the input tensor $\mathcal{M}_{\text{2D}}$ is first processed by the adapter block to extract spatio-temporal features, which are encoded into latent sequence $\tilde{\mathbf{a}}_{1:u} \in \mathbb{R}^{u \times d}$ with $u = F/\alpha$ ($\alpha$: down-sampling rate) and latent dimension $d$. This sequence is quantized via $N + 1$ residual vector quanti-zation (RVQ) layers (Guo et al., 2024; Zeghidour et al., 2021). The base layer encodes the primary latent $\tilde{\mathbf{a}}_{1:u}$, while subsequent layers iteratively refine residuals. At each layer $n$, the residual $\mathbf{r}^n$ is quantized by replacing it with the nearest codebook entry from a learnable codebook $\mathcal{C}$ where each codebook entry $\mathbf{c}_k$ is referred to as a motion token, representing a discrete motion pattern learned from the training data. Formally, the quantization function $Q(\cdot)$ is defined as:

$$\mathbf{a}^n = Q(\mathbf{r}^n) = \arg\min_{\mathbf{c}_k \in \mathcal{C}} \|\mathbf{r}^n - \mathbf{c}_k\|_2, \quad \mathbf{r}^{n+1} = \mathbf{r}^n - \mathbf{a}^n, \tag{1}$$

with the initial residual given by $\mathbf{r}^0 = \tilde{\mathbf{a}}_{1:u}$. The final quantized representation $\mathbf{a}^q = \sum_{n=0}^{N} \mathbf{a}^n$ is decoded into SMPL parameters $\mathcal{M}_{\text{SMPL}}$.

The overall training objective is the latent embedding loss in each quantization layer:

$$\mathcal{L}_{\text{rvq}} = \lambda_{\text{rvq}} \sum_{n=1}^{N} \|\mathbf{r}^n - \text{sg}[\mathbf{a}^n]\|_2^2, \tag{2}$$

where $\text{sg}[\cdot]$ denotes the stop-gradient operation and $\lambda_{\text{rvq}}$ is a weighting factor.

**3D-to-2D Projection Model.** In order to calculate the loss between $\mathcal{M}_{\text{SMPL}}$ and $\mathcal{M}_{\text{2D}}$, we employ a 3D-to-2D projection model, which transforms the generated SMPL motion sequences into 2D keypoints. Specifically, given the generated SMPL parameters, the SMPL model (Pavlakos et al., 2019) reconstructs a 3D human mesh with 6,890 vertices. We project the mesh onto the 2D image plane and then extract keypoints corresponding to the COCO format (Lin et al., 2014) as our skeletal representation. The SMPL model is fully differentiable, making it compatible with AI training frameworks.

For each 3D joint position $\mathbf{p}_i = (x_i, y_i, z_i)$ generated by SMPL model, the corresponding 2D coordinates $\mathbf{k}_i = (u_i, v_i)$ on the image plane are obtained via:

$$u_i = \frac{f \cdot x_i}{z_i + \epsilon} + \frac{W}{2}, \quad v_i = \frac{f \cdot y_i}{z_i + \epsilon} + \frac{H}{2}, \tag{3}$$

where $f$ is the focal length of the camera model, set to 5,000 in this case, $(W, H)$ denotes the original image width and height, and $\epsilon$ is a small constant to avoid division by zero. And to ensure numerical stability, we filter out the joints where $z_i \leq 0$, as these correspond to the points behind the camera. After projection, we extract a subset of keypoints following the COCO format to form the final 2D keypoint representation $\mathbf{M}_{2D} \in \mathbb{R}^{F \times J \times 2}$, where $F$ is the number of frames, $J$ is the number of keypoints, and each keypoint is described by its 2D coordinates $(x, y)$.

This approach ensures efficient 3D-to-2D projection by filtering out invalid projections caused by depth inconsistencies, enabling the effective use of 2D supervision.

### 3.3 3D-FREE SUPERVISION

Our method is fully 3D-free, requiring no 3D annotations. Supervision is achieved solely through 2D keypoints, with geometric losses and structural constraints ensuring robust and accurate results.

**Geometric Loss.** We denote the ground truth 2D keypoints sequence as $\mathbf{M}_{2D}^{\text{true}}$ and the predicted keypoints sequence as $\mathbf{M}_{2D}^{\text{pred}}$. Furthermore, a confidence mask $\mathbf{M}_{\text{mask}} \in \mathbb{R}^{F \times J}$ is introduced to mask out keypoints with confidence scores below a threshold (e.g., 0.8), ensuring unreliable keypoints are excluded from loss computation. Apart from $\mathcal{L}_{\text{rvq}}$, we design several geometric losses to enforce physical properties and promote natural and coherent motion.

The motion reconstruction loss ensures frame-wise consistency between the predicted and ground truth sequences by enforcing per-frame similarity:

$$\mathcal{L}_{\text{rec}} = \left\| \mathbf{M}_{\text{mask}} \odot \mathbf{M}_{2D}^{\text{true}} - \mathbf{M}_{\text{mask}} \odot \mathbf{M}_{2D}^{\text{pred}} \right\|_1. \tag{4}$$

To preserve motion dynamics, the velocity loss penalizes discrepancies in temporal changes by comparing velocity differences frame by frame:

$$\mathcal{L}_{\text{vel}} = \sum_{t=1}^{N} \left\| \mathbf{M}_{\text{mask}}^t \odot (\mathbf{M}_{2D}^{\text{true},t} - \mathbf{M}_{2D}^{\text{true},t-1}) - \mathbf{M}_{\text{mask}}^t \odot (\mathbf{M}_{2D}^{\text{pred},t} - \mathbf{M}_{2D}^{\text{pred},t-1}) \right\|_1, \tag{5}$$

where $t \in \{0, 1, \ldots, F-1\}, \quad t \in \mathbb{Z}$ and $\mathbf{M}_{\text{mask}}^t \in \mathbb{R}^{(F-1) \times J}$ ensures that keypoint at time $t$ is preserved only if the confidence scores in both the current and previous frames exceed the threshold, preventing unreliable keypoints from being penalized.

To encourage temporal smoothness, we introduce a loss that suppresses sudden changes in predicted motion between consecutive frames:

$$\mathcal{L}_{\text{smooth}} = \sum_{t=1}^{N} \left\| \mathbf{M}_{2D}^{\text{pred},t} - \mathbf{M}_{2D}^{\text{pred},t-1} \right\|_1. \tag{6}$$

Besides, during training, we found that the generated SMPL parameters fluctuate significantly at the boundaries. Therefore, to make the motion smoother and more realistic, we impose additional constraints on the first and last frames:

$$\mathcal{L}_{\text{boundary}} = \left\| \mathbf{M}_{\text{mask}} \odot (\mathbf{M}_{2D}^{\text{true},N-1} - \mathbf{M}_{2D}^{\text{pred},N-1}) \right\|_1 + \left\| \mathbf{M}_{\text{mask}} \odot (\mathbf{M}_{2D}^{\text{true},0} - \mathbf{M}_{2D}^{\text{pred},0}) \right\|_1. \tag{7}$$

**Pose and Shape Constraints.** Furthermore, to enhance the quality of the generated motion sequences, we introduce additional constraints. These additional losses serve to improve the realism, smoothness, and consistency of the generated motions, ensuring alignment with human-perceived natural movement.

When projecting generated motion onto a 2D plane, multiple SMPL parameter configurations can match the keypoints, but many are physically implausible. These may result in unrealistic poses, including severe body self-intersections or anatomically impossible joint rotations. Unconstrained generation may also cause significant z-axis oscillations.

To mitigate these issues, we impose **constraints** on pose parameters $\boldsymbol{\theta}$ to ensure the generation of physically feasible 3D human motions. We analyze the AMASS dataset (Mahmood et al., 2019) to statistically define valid pose parameter ranges and refine them through visualization tests, adjusting thresholds to remove unnatural poses. This constraint is enforced via the loss function:

$$\mathcal{L}_{\text{pose}} = (\max(0, \theta - \theta_{\max}) + \max(0, \theta_{\min} - \theta)). \tag{8}$$

Table 1: Comparisons between **MotionWeb** and existing 3D text-motion datasets.

| Dataset | # Clip | Hour | Source | # Caption | Indoor | Outdoor | RGB |
|---|---|---|---|---|---|---|---|
| KIT-ML (Plappert et al., 2016) | 3,911 | 11.2 | Marker-based MoCap | 6,278 | ✓ | ✗ | ✗ |
| AMASS (Mahmood et al., 2019) | 11,265 | 40.0 | Marker-based MoCap | N/A | ✓ | ✗ | ✗ |
| BABEL (Punnakkal et al., 2021) | 13,220 | 43.5 | Marker-based MoCap | 91,408 | ✓ | ✗ | ✗ |
| HumanML3D (Guo et al., 2022a) | 14,616 | 28.6 | Marker-based MoCap | 44,970 | ✓ | ✗ | ✗ |
| SnapMoGen (Hwang et al.) | 20,450 | 43.7 | Marker-based MoCap | 122,565 | ✓ | ✗ | ✗ |
| Motion-X (Lin et al., 2023) | 81,084 | 144.2 | Pseudo GT & MoCap | 81,084 | ✓ | ✓ | ✓ |
| **MotionWeb** (Ours) | 101,208 | 160.2 | Pseudo GT | 154,680 | ✓ | ✓ | ✓ |

Similarly, we introduce constraints on the shape parameters $\boldsymbol{\beta}$ to prevent unrealistic body proportions. These restrictions ensure that the generated human models maintain a natural and plausible appearance, improving both the visual quality and physical accuracy of the synthesized motions. To achieve this, we minimize deviations from a statistically valid range:

$$\mathcal{L}_{\text{shape}} = \max(0, 0.1 \sum_{i=1}^{10} |\beta_i| - \tau), \tag{9}$$

where $\tau$ is the predefined threshold that restricts the body shape within realistic limits.

Finally, the total loss is defined as:

$$\mathcal{L} = \mathcal{L}_{\text{rvq}} + \lambda_{\text{rec}}\mathcal{L}_{\text{rec}} + \lambda_{\text{smooth}}\mathcal{L}_{\text{smooth}} + \lambda_{\text{vel}}\mathcal{L}_{\text{vel}} + \lambda_{\text{boundary}}\mathcal{L}_{\text{boundary}} + \lambda_{\text{pose}}\mathcal{L}_{\text{pose}} + \lambda_{\text{shape}}\mathcal{L}_{\text{shape}}, \tag{10}$$

where $\lambda_{\text{rec}}, \lambda_{\text{smooth}}, \lambda_{\text{vel}}, \lambda_{\text{boundary}}, \lambda_{\text{pose}}, \lambda_{\text{shape}}$ are hyperparameters that control the relative importance of each term. The specific numerical settings can be found in Appendix A.4.

### 3.4 TEXT-TO-MOTION TRANSFORMER

As shown in Figure 2(b), our training framework employs two text-conditioned transformers. The goal is to predict pre-trained motion tokens based on textual descriptions **T**. The first component is the masked transformer, which reconstructs masked motion tokens $\tilde{m}^0 \in \mathbb{R}^u$ at the base layer. Specifically, a portion of the original base layer motion tokens $m^0_{1:u} \in \mathbb{R}^u$ is randomly masked by replacing selected tokens with a special [MASK] token, following the approach in (Devlin et al., 2019). The masked transformer $f_m$ then predicts the masked tokens by conditioning on the partially observed sequence $\tilde{m}^0$ and the corresponding text features $\mathbf{c}$ extracted from CLIP (Radford et al., 2021). This enables the model to learn robust representations by reconstructing the missing motion tokens based on both the observed motion context and the semantic guidance from the text. The model is trained by minimizing the negative log-likelihood of the target predictions:

$$\mathcal{L}_{\text{mask}} = \sum_{\tilde{m}^0_i = [\text{MASK}]} -\log f_m\left(m^0_i \mid \tilde{m}^0, \mathbf{c}\right). \tag{11}$$

In parallel, a residual transformer is trained to predict tokens across $N$ residual quantization layers. For each residual layer $j \in [1, N]$, the tokens from all preceding layers $m^{0:j-1}$ are embedded and summed to form the input representation. Conditioned on this token embedding, the text features $\mathbf{c}$, and residual layer indicator $j$, the residual transformer $f_r$ predicts the $j$-th layer tokens $m_j$, with the following training objective:

$$\mathcal{L}_{\text{res}} = \sum_{j=1}^{N} \sum_{i=1}^{u} -\log f_r\left(m^j_i \mid m^{0:j-1}_i, \mathbf{c}, j\right). \tag{12}$$

**Inference.** The inference process is shown in Figure 2 with dashed line. The generation starts from a fully masked token sequence, guided by the textual input. The base-layer token sequence is refined iteratively. At each step, the masked transformer predicts tokens for masked positions, re-masking the least confident ones until the sequence is fully determined. The residual transformer then predicts the tokens for the subsequent $N$ residual quantization layers, progressively refining motion details at each layer. After predicting all layers, the full quantized motion token sequence is obtained. The final motion embedding is retrieved from the pre-trained codebook and passed to the MotionVQ-Adapter decoder to reconstruct the coherent SMPL motion sequence.

## 4 MotionWeb Dataset

We introduce **MotionWeb**, a large-scale dataset containing more than 100k motion clips extracted from videos. The quantitative comparisons of **MotionWeb** and existing datasets in Table 1.

**Data Collection.** The data collection pipeline mainly consists of using motion keywords to collect videos from public platforms like YouTube, extracting keypoints, trimming clips, and generating captions. Since the motion in the video typically combines both camera movement and human motion, we aim to remove the camera movement and focus solely on human actions. To achieve this, a filtering method is applied to **exclude** clips with **camera motion**, ensuring consistency in the global translation $t$ of the SMPL model. Furthermore, we designed task-specific prompts for the video captioning model to ensure accurate motion description generation within our pipeline. More details of these methods are provided in Appendix A.2.

The **MotionWeb** dataset consists of video data collected from two primary sources. The first part comprises a set of videos retrieved from the Internet using 112 distinct keywords, resulting in a total of 17,796 monocular videos spanning over 2,583 hours. The second part includes video samples from the IDEA400 (Lin et al., 2023) and GRAB dataset (Taheri et al., 2020), which provides additional motion diversity and scene variations. By combining these sources, the **MotionWeb** dataset offers a broad spectrum of human motion scenarios, facilitating the training of motion generation models under various conditions.

Our dataset consists of textual descriptions paired with 2D keypoint sequences, represented as $\mathcal{M}_{2D} \in \mathbb{R}^{F \times 54}$. The dimension 54 is derived from 18 keypoints, each described by 3 values: $(x, y)$ pixel coordinates and a confidence score. This format aligns with the COCO standard (Lin et al., 2014), which defines 17 human keypoints. We extend this by adding an 18th keypoint to store the width and height of the original video, with its confidence score fixed at 1 across all frames. This facilitates precise 3D-to-2D mapping by providing the necessary spatial context. Each motion clip in **MotionWeb** has one or two textual annotations, sampled at 30 fps, with each clip lasting between 2 to 10 seconds. In total, the dataset consists of 101,208 motion sequences and 154,680 textual descriptions. More details on our dataset statistics are provided in Appendix A.2.

## 5 Experiments

### 5.1 Implementation Detail

Our framework is trained on an NVIDIA A40 GPU with PyTorch. For training MotionVQ-Adapter, the batch size is set to 256 and the learning rate is set to 2e-4. The data is then represented by the joint VQ codebook comprised of 512 codes, each with a dimension of 512. For the residual structure, the number of residual quantization layers is set to 6 following the previous method (Guo et al., 2024). For the adapter, we use a DSTformer with depth 1, 8 heads, a feature size of 512, and an embedding size of 512. The masked transformer and the residual transfomer are set to have 6 transformer layers, 6 heads, and 384 latent dimensions. For transformer training, the batch size is set to 64 and the learning rate is set to 2e-4. We use the Adam optimizer (Kingma, 2014) for training.

### 5.2 Benchmark and Metrics

**HumanML3D.** The HumanML3D dataset (Guo et al., 2022a), a standard benchmark comprising 14,616 motion sequences (sourced from AMASS (Mahmood et al., 2019) and HumanAct12 (Guo et al., 2020)), is extensively employed in 3D motion generation research. For our experiments, we derive 2D keypoints from AMASS 3D motion visualizations, preserve the original dataset partitions (train/validation/test), and utilize the processed data for model training and evaluation.

Due to differences between the DMPL model (Loper et al., 2015) used in AMASS and SMPL (different parameter dimensions) and the occasional lack of translation information in AMASS visualizations, we cannot convert the SMPL parameters into the 263-dimensional representation (Guo et al., 2022a) used in HumanML3D. Therefore, traditional evaluation metrics are not applicable. Instead, we use the *joint2smpl* method (Zuo et al., 2021) to transform the results of other SOTA methods into SMPL parameters and evaluate motion quality with MotionCritic (Wang et al., 2024a). Motion-

Table 2: Comparison of MotionCritic scores on different methods. **Bold** indicates the best result.

| Method | Ave. Score ($\uparrow$) | Max ($\uparrow$) | Min ($\uparrow$) | Std ($\downarrow$) |
|---|---|---|---|---|
| MoMask (Guo et al., 2024) | -2.283 | 4.518 | -14.308 | 3.475 |
| CrossDiff (Ren et al., 2024) | -2.155 | 4.320 | -13.950 | 3.360 |
| MDM (Tevet et al., 2022b) | -2.094 | 4.137 | -10.867 | 3.022 |
| MLD (Chen et al., 2023) | -2.013 | 6.052 | -13.583 | 3.297 |
| MMM (Pinyoanuntapong et al., 2024) | -2.010 | 7.076 | -17.129 | 3.438 |
| MoMask++ (Hwang et al.) | -1.925 | 5.110 | -12.450 | 3.310 |
| ReMoDiffuse (Zhang et al., 2023b) | -1.860 | 5.433 | -13.219 | 3.381 |
| MotionGPT3 (Zhu et al., 2025) | -1.745 | 5.920 | -12.760 | 3.250 |
| T2M-GPT (Zhang et al., 2023a) | -1.709 | 7.062 | -12.321 | 3.402 |
| T2M (Guo et al., 2022a) | -1.659 | 4.073 | -14.213 | 3.116 |
| MotionDiffuse (Zhang et al., 2024a) | -1.533 | 6.990 | -11.910 | 3.196 |
| mogents (Yuan et al., 2024) | -1.420 | 6.850 | -10.150 | 2.980 |
| ReMoMask (Li et al., 2025) | -1.150 | 7.340 | -9.230 | 2.850 |
| **K2M** (Ours) | **-0.623** | **7.867** | **-7.614** | **2.530** |

Table 3: **Comparison with the state-of-the-art methods on MotionWeb dataset.** The right arrow $\rightarrow$ means the closer to real motion the better. **Bold** indicates the best result.

| Method | FID ($\downarrow$) | Top@1 ($\uparrow$) | Top@2 ($\uparrow$) | Top@3 ($\uparrow$) | MM-Dist ($\downarrow$) | Diversity ($\rightarrow$) |
|---|---|---|---|---|---|---|
| Ground Truth | $0.002^{\pm0.000}$ | $0.538^{\pm0.003}$ | $0.722^{\pm0.003}$ | $0.808^{\pm0.002}$ | $3.327^{\pm0.008}$ | $9.688^{\pm0.065}$ |
| T2M (Guo et al., 2022a) | $9.845^{\pm0.017}$ | $0.202^{\pm0.001}$ | $0.327^{\pm0.001}$ | $0.417^{\pm0.001}$ | $6.166^{\pm0.005}$ | $6.959^{\pm0.049}$ |
| MoMask (Guo et al., 2024) | $5.590^{\pm0.021}$ | $0.340^{\pm0.001}$ | $0.521^{\pm0.002}$ | $0.633^{\pm0.001}$ | $4.652^{\pm0.005}$ | $8.181^{\pm0.085}$ |
| MotionGPT3 (Zhu et al., 2025) | $4.720^{\pm0.019}$ | $0.365^{\pm0.002}$ | $0.545^{\pm0.002}$ | $0.650^{\pm0.001}$ | $4.590^{\pm0.006}$ | $8.320^{\pm0.078}$ |
| MoMask++ (Hwang et al.) | $5.150^{\pm0.020}$ | $0.352^{\pm0.002}$ | $0.533^{\pm0.001}$ | $0.641^{\pm0.002}$ | $4.620^{\pm0.006}$ | $8.250^{\pm0.080}$ |
| mogents (Yuan et al., 2024) | $3.850^{\pm0.016}$ | $0.382^{\pm0.002}$ | $0.560^{\pm0.001}$ | $0.665^{\pm0.002}$ | $4.480^{\pm0.005}$ | $8.450^{\pm0.065}$ |
| MMM (Pinyoanuntapong et al., 2024) | $2.732^{\pm0.018}$ | $0.299^{\pm0.002}$ | $0.444^{\pm0.001}$ | $0.541^{\pm0.001}$ | $4.958^{\pm0.006}$ | $8.102^{\pm0.057}$ |
| ReMoMask (Li et al., 2025) | $2.685^{\pm0.015}$ | $0.395^{\pm0.001}$ | $0.575^{\pm0.001}$ | $0.678^{\pm0.001}$ | $4.350^{\pm0.004}$ | $8.650^{\pm0.072}$ |
| **K2M**(Ours) | $\mathbf{2.638}^{\pm0.014}$ | $\mathbf{0.415}^{\pm0.002}$ | $\mathbf{0.594}^{\pm0.001}$ | $\mathbf{0.694}^{\pm0.001}$ | $\mathbf{4.191}^{\pm0.005}$ | $\mathbf{8.831}^{\pm0.092}$ |

Critic scores motion sequences directly from SMPL parameters, assessing naturalness, smoothness, and realism, with higher scores indicating better quality. It outperforms previous metrics in aligning with human perceptions and generalizes well across data distributions, serving as an effective motion quality metric. (Wang et al., 2024a)

**MotionWeb.** On **MotionWeb**, we adopt common evaluation metrics from prior 3D human motion generation studies (Guo et al., 2022a; Tevet et al., 2022b; Zhang et al., 2023a; Guo et al., 2024) to assess our framework at the 2D level, including Frechet Inception Distance (FID), R-Precision, Multimodal Distance (MM-Dist), and Diversity. More details are in Appendix A.3. To compute these metrics, we retrain the evaluator from (Guo et al., 2022a) using our dataset.

### 5.3 RESULT

**Quantitative results.** To evaluate **K2M** on the HumanML3D, we benchmark it against various SOTA methods, including VAE (Guo et al., 2022a), autoregressive models (Zhang et al., 2023a; Guo et al., 2024; Pinyoanuntapong et al., 2024), and diffusion-based models (Tevet et al., 2022b; Chen et al., 2023; Zhang et al., 2024a; 2023b). We randomly sample 500 text descriptions from the test set, generate corresponding motion sequences, and assess them using MotionCritic. As shown in Table 2, **K2M** achieves the highest mean, maximum, and minimum values while maintaining the lowest standard deviation, demonstrating its ability to generate natural and human-judgment-consistent SMPL motion sequences.

The quantitative results for **MotionWeb** are shown in Table 3. We primarily selected several VAE-based (Guo et al., 2022a) and VQ-based methods (Pinyoanuntapong et al., 2024; Guo et al., 2024) for comparison, as diffusion models are not suitable for the scenario where the input and output dimensions differ. We modified the output dimensions of these methods, employing the 3D-to-2D projection model for 2D supervision and applying the same geometric loss and constraints. In line with previous works (Pinyoanuntapong et al., 2024; Guo et al., 2024), all experiments were conducted 20 times, with results presented as mean values accompanied by 95% confidence intervals. In this scenario, **K2M** achieves state-of-the-art performance across all metrics, with a significant

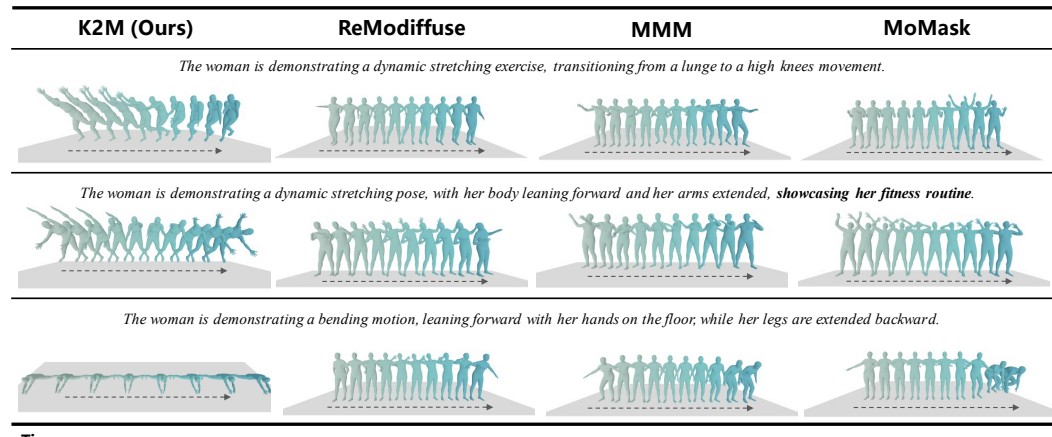

Figure 3: **Qualitative comparisons on the test set of MotionWeb.** The visualization results are arranged in chronological order. Only key frames are displayed.

lead over other methods in terms of FID and R-precision. This indicates that our method not only generates realistic motions but also ensures they align well with the input textual information.

Then we explored a pre-training strategy where our motion prior was first trained on the GRAB dataset (Taheri et al., 2020) before being fine-tuned on our MotionWeb dataset. Our goal was to leverage a high-quality 3D MoCap dataset to pre-train our motion prior. The GRAB dataset was specifically chosen because it includes full video data, allowing us to extract the 2D keypoints compatible with our training pipeline. This compatibility is not feasible with other popular datasets as their visualizations often lack global translation. We compare the performance of this pre-trained model against our original model and the Ground Truth in Table 4. The results show that pre-training leads to an improvement in the FID score and R-Precision, which aligns with our qualitative observations of reduced noise and jitter in the resulting motion visualizations. However, we also observed that the overall benefit was constrained by the limited scale and diversity of the GRAB dataset.

Table 4: Quantitative results for the pre-training experiment.

| Model | FID ↓ | R-Precision @ Top 1 ↑ | @ Top 2 ↑ | @ Top 3 ↑ |
|---|---|---|---|---|
| Ground Truth | 0.002 | 0.538 | 0.722 | 0.808 |
| K2M (Ours) | 2.638 | 0.415 | 0.594 | 0.694 |
| K2M + Pre-training | **2.622** | **0.421** | **0.602** | **0.691** |

**Qualitative comparisons.** Figure 3 presents qualitative comparisons between our approach and SOTA methods, including ReMoDiffuse (Zhang et al., 2023b), MoMask (Guo et al., 2024), and MMM (Pinyoanuntapong et al., 2024). The results show that other methods often miss key details for complex motions and fail entirely for outdoor actions. In contrast, **K2M** generates text-aligned motions, including outdoor scenarios. Besides, **K2M** directly generates SMPL parameters, eliminating intermediate conversions. When processing outputs from traditional 3D human motion generation models, *joint2smpl* conversion Zuo et al. (2021) is required to transform joint-based representations into SMPL parameters before visualization. In contrast, **K2M** directly generates SMPL parameters, eliminating this conversion step. Based on statistics from visualizing 20 samples, the *joint2smpl* conversion takes an average of 0.672864 seconds per frame, while visualization requires 0.229749 seconds per frame. Thus, by using our output format, the entire visualization process achieves approximately 75% higher efficiency. Furthermore, if the generated SMPL format is directly used in subsequent applications, our model further enhances computational efficiency.

In addition, we conducted a user study to evaluate **K2M** against SOTA methods via side-by-side comparison with 35 participants, generating 40 motions per method from **MotionWeb** test texts. As shown in Figure 10, **K2M** was preferred over competing models in most cases.

Table 5: Comparative evaluation of motion generation methods.

(a) Ablation study of loss components.

| Removed Objective | Recon. Loss | $\Delta$ Drop |
|---|---|---|
| None (Full Model) | 6.693 | – |
| $-\mathcal{L}_{\text{smooth}}$ | 7.035 | 0.342 |
| $-\mathcal{L}_{\text{vel}}$ | 6.931 | 0.238 |
| $-\mathcal{L}_{\text{boundary}}$ | 8.136 | 1.443 |

(b) Motion VQ designs comparison.

| Methods | FID ($\downarrow$) | Top@1 ($\uparrow$) |
|---|---|---|
| T2M-GPT (Guo et al., 2024) | $9.687^{\pm 0.005}$ | $0.266^{\pm 0.001}$ |
| MMM (Pinyoanuntapong et al., 2024) | $1.632^{\pm 0.004}$ | $0.391^{\pm 0.001}$ |
| MoMask (Guo et al., 2024) | $0.329^{\pm 0.004}$ | $0.493^{\pm 0.001}$ |
| **K2M** (Ours) | $\mathbf{0.085}^{\pm \mathbf{0.001}}$ | $\mathbf{0.511}^{\pm \mathbf{0.001}}$ |

## 5.4 ABLATION STUDY

**Adapter.** We evaluate motion quantization using two metrics: FID and R-Precision Top-1 following previous methods (Zhang et al., 2023a; Guo et al., 2024; Pinyoanuntapong et al., 2024). To validate MotionVQ-Adapter, we first train the original residual VQ-VAE on the **MotionWeb** dataset, then integrate our adapter and compare performance. Additionally, we benchmark against other VQ-based motion generation methods. As shown in Table 5b, our approach outperforms all baselines across all metrics.

**Losses and Constraints.** To validate our designed losses, we conduct an ablation study on 10k samples, excluding $\mathcal{L}_{\text{rvq}}$ and $\mathcal{L}_{\text{rec}}$ as essential. Using converged $\mathcal{L}_{\text{rec}}$ as the metric, Table 5a shows removing any loss degrades performance. While pose and shape constraints do not affect numerical metrics, their impact is clear visually. Figure 9 shows unnatural twisting in the upper body and left arm without pose constraints (Figure 9(b)), and body shape deviates from human proportions without shape constraints (Figure 9(d)).

## 6 CONCLUSION

In this paper, we presented **K2M**, a novel text-driven 3D human motion generation approach that leverages 3D-free supervision from web videos, alongside **MotionWeb**, a large-scale dataset of 2D motion clips paired with textual captions offering unprecedented diversity for training. **K2M** not only achieves superior performance across multiple metrics compared to existing methods, but it also overcomes a significant technical barrier by eliminating the traditional reliance on MoCap lab environments. By demonstrating the feasibility of directly generating high-quality 3D motion sequences with only 2D supervision, our work simplifies the data acquisition process and empowers researchers and practitioners to construct their own training datasets and models, thus democratizing access to advanced motion synthesis technology. We anticipate that these contributions will advance research in 3D human motion generation, paving the way for more robust and versatile text-driven motion synthesis systems.

**Limitations and future work.** Like most other 3D human motion generation methods, our model requires the target length as input. This issue can be effectively addressed by applying a length prediction model (Guo et al., 2022a). Our 2D-derived motion data, while computationally efficient to produce, exhibits minor inconsistencies such as occasional jitter from pose estimation. This represents a deliberate trade-off compared to traditional marker-based MoCap systems that require expensive equipment and labor-intensive annotation. Nevertheless, our quantitative evaluations demonstrate that the proposed method achieves competitive motion quality compared to fully supervised 3D approaches. The main technical limitation lies in the weaker constraints on depth (z-axis) estimation. To address current limitations, future work will focus on: (1) incorporating monocular depth estimation to improve 3D consistency, and (2) leveraging additional motion datasets to enhance generalization.

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

## A APPENDIX

The appendix material is structured as follows:

- Case study in Section A.1.
- Additional dataset collection pipeline and statistics in Section A.2.
- More details on the evaluation metrics are given in Section A.3.
- More details about our experiment in Section A.4.
- The details of using Large Language Models (LLMs) are given in Section A.5.

### A.1 CASE STUDY

Figure 4 presents additional qualitative results of **K2M**. Based on the results, our method is capable of generating complex 3D motions that current methods are unable to produce, with significant improvements in both motion intensity and diversity.

We have tackled two major challenges faced by current 3D human motion generation methods: first, the diversity of the generated motions has significantly improved, allowing us to generate more diverse and outdoor motions; second, we are able to generate more intense movements, thanks to the inherent characteristics of the SMPL parameters.

And here I want to discuss the societal impacts of the work. Our work presents positive societal impacts by democratizing 3D motion generation through a method trainable on simple, self-collected 2D datasets, lowering the technical and financial barriers associated with traditional motion capture systems. This advancement could benefit applications in animation, virtual reality, and biomechanics research. However, we acknowledge the potential negative impact that generated 3D motions could be misused to create synthetic animations for malicious purposes, such as producing deepfake content for impersonation or disinformation. While our current work focuses on technical advancement rather than specific applications, we emphasize the importance of ethical considerations and responsible use as this technology develops.

### A.2 MOTIONWEB COLLECTION AND STATISTICS

The detailed data information of **MotionWeb** dataset is in Table 6. The number of video clips we collected from the Web is 75,307, which is approximately 74.4% of the total. The number of clips sourced from other video datasets is 25,901, representing about 25.6% of the total. The video datasets include IDEA400 Lin et al. (2023) and GRAB dataset Taheri et al. (2020). IDEA400, a subset of Motion-X Lin et al. (2023), was collected by their team and consists of 400 actions. The original videos are available for download. Additionally, the GRAB dataset is a motion capture dataset featuring interactions with 51 everyday objects and provides visualization results.

Table 6: **MotionWeb dataset statistics.**

| Attribute | Value |
|---|---|
| Motion clips | 101,208 |
| Descriptions | 154,680 |
| Unique words | 6,699 |
| Total duration | 160.2 h |
| Total frames | 17,297,263 |
| Avg. motion length | 5.697 s |
| Avg. description length | 21.61 words |

Furthermore, we also explore the joint diversity and strength distribution of **MotionWeb** dataset. Figure 6 presents the average velocity of each keypoint in COCO format Lin et al. (2014). We observe that the velocity is highest at the elbow, wrist, knee, and ankle, which aligns with the expected behavior. Overall, the velocities are relatively high, indicating that the motion intensity throughout the dataset is strong.

As illustrated in Figure 5, the overall data collection pipeline of our **MotionWeb** dataset consists of the following steps: 1) generating motion keywords using a large language model (LLM) Brown

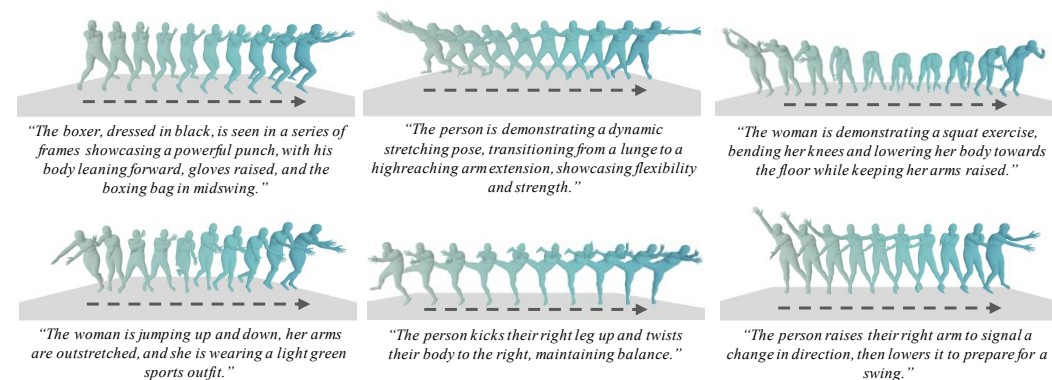

*"The boxer, dressed in black, is seen in a series of frames showcasing a powerful punch, with his body leaning forward, gloves raised, and the boxing bag in midswing."*

*"The person is demonstrating a dynamic stretching pose, transitioning from a lunge to a highreaching arm extension, showcasing flexibility and strength."*

*"The woman is demonstrating a squat exercise, bending her knees and lowering her body towards the floor while keeping her arms raised."*

*"The woman is jumping up and down, her arms are outstretched, and she is wearing a light green sports outfit."*

*"The person kicks their right leg up and twists their body to the right, maintaining balance."*

*"The person raises their right arm to signal a change in direction, then lowers it to prepare for a swing."*

Figure 4: **More qualitative results of our method.** The visualization results are arranged in chronological order. Only key frames are displayed.

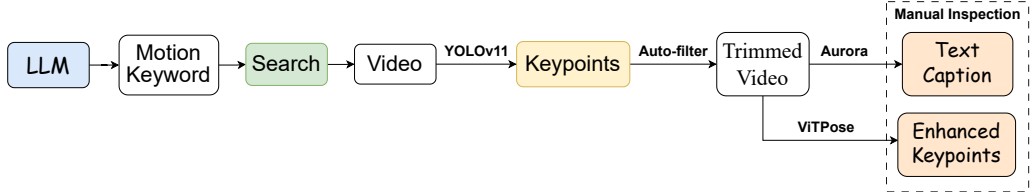

Figure 5: Diagram depicting the complete process of data collection and annotation.

et al. (2020), 2) collecting videos from publicly available sources such as YouTube and Bilibili, 3) extracting keypoints using the YOLOv11 pose model Jocher et al. (2022) for its high speed, preserving the information of the largest detection box in each frame, filtering motion sequences based on keypoint information, and trimming video clips accordingly, 4) re-extracting keypoints from the trimmed video clips using ViTPose Xu et al. (2022) for enhanced accuracy, 5) generating textual descriptions with video captioning model Chai et al. (2024), and 6) performing manual inspection to ensure data quality.

Additionally, I would like to supplement some details regarding the pipeline. First, we employ the Vitpose-H model, which offers the highest precision among available pose estimation models. During the trimming of video clips using the YOLOv11 pose model, we save the visualization results and annotate the target person with a red bounding box. This facilitates the subsequent step of adding textual annotations. Consequently, when utilizing the video captioning model, our prompt is: "Use one or two concise sentences to describe the motion of the person within the red bounding box in chronological order, without referencing their physical appearance." This prompt formulation explicitly constrains the captioning model to focus on kinematic descriptions while eliminating confounding variables.

The filtering criteria are as follows:

1) Camera movement detection: We evaluate pixel changes at the left and right edges of the video to determine if the camera has shifted. If movement is detected, the corresponding frames are removed. 2) Speed filtering ensures that the average velocity and acceleration of keypoints do not exceed a maximum threshold to prevent abrupt motion or teleportation. Additionally, keypoint speeds should not remain below a threshold to avoid the model from generating completely stationary poses. 3) Confidence-based filtering ensures that more than 75% of the keypoints have confidence scores above a predefined threshold, and no keypoints below the pelvis should have confidence scores lower than this threshold. 4) The bounding box height of the human body in the image should occupy more than one-third of the image height to ensure reliable detection.

Examples from the MotionWeb dataset are illustrated in Figure 7. The left half of the figure displays 10 keyframes extracted from a video, along with their corresponding 2D keypoint annotations and

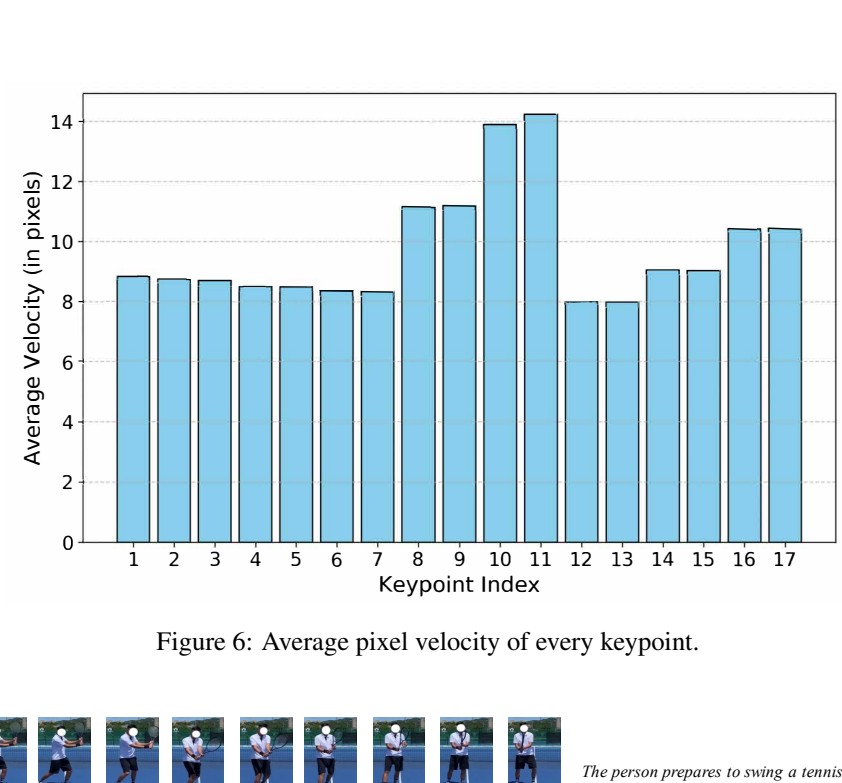

Figure 6: Average pixel velocity of every keypoint.

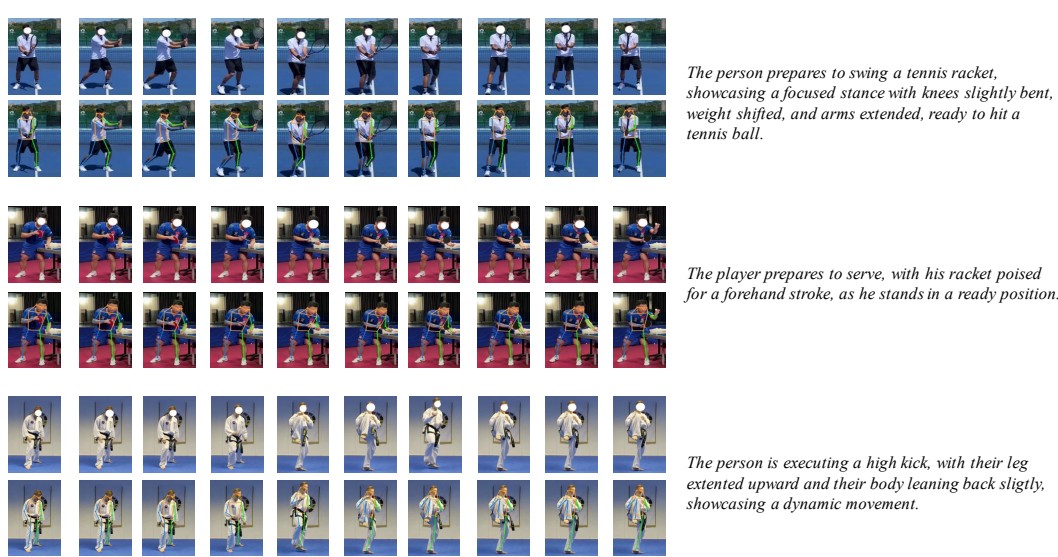

*The person prepares to swing a tennis racket, showcasing a focused stance with knees slightly bent, weight shifted, and arms extended, ready to hit a tennis ball.*

*The player prepares to serve, with his racket poised for a forehand stroke, as he stands in a ready position.*

*The person is executing a high kick, with their leg extented upward and their body leaning back sligtly, showcasing a dynamic movement.*

Figure 7: Showcase of data examples in **MotionWeb**.

visualizations. The right half presents the textual descriptions generated by the video captioning model based on the video content. As shown, the 2D keypoint annotations are highly accurate, and the textual descriptions are both detailed and precise, demonstrating the quality of our data pipeline.

Table 7: Viewpoint Diversity Statistics for MotionWeb Dataset

| Viewpoint | Frame Count | Percentage |
|---|---|---|
| Front | 6,816,810 | 39.41% |
| Left Side | 4,524,532 | 26.16% |
| Right Side | 4,726,061 | 27.32% |
| Back | 1,229,860 | 7.11% |
| **Total** | **17,297,263** | **100.00%** |

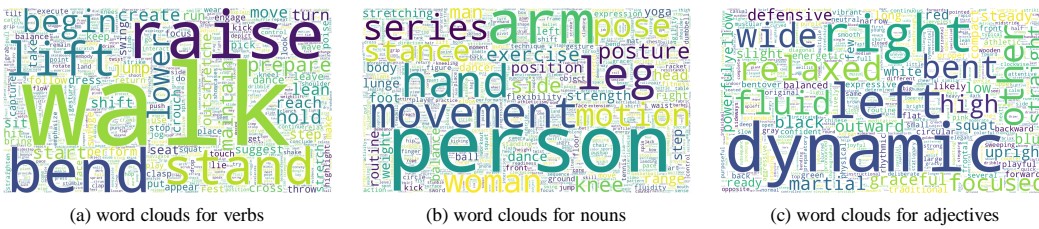

(a) word clouds for verbs        (b) word clouds for nouns        (c) word clouds for adjectives

Figure 8: Word clouds for the annotated text of **MotionWeb**.

To demonstrate the diversity of our dataset, we generated word clouds for verbs, nouns, and adjectives extracted from the annotated text, as illustrated in Figure 8. As shown in the figure, the text annotations exhibit a rich variety across all parts of speech, highlighting the extensive diversity of our dataset. This richness not only underscores the comprehensiveness of our annotations but also ensures that the dataset can support a wide range of motion generation tasks. Furthermore, to quantify the spatial diversity captured in our data, we provide statistics on viewpoint distribution for our MotionWeb dataset in Table 7. As these statistics demonstrate, MotionWeb is highly diverse and not dominated by a single viewpoint, with significant contributions from front, left, right, and even back perspectives. This multi-perspective information, aggregated from different sequences, is fundamental to resolving ambiguities and learning a generalizable 3D motion prior, thereby enhancing the robustness and applicability of our method.

We will release the keypoint information and textual annotations of **MotionWeb** dataset. Due to license and copyright restrictions, we will not directly release the original video clips. However, we will provide download links for each video.

### A.3 EVALUATION METRICS

Here we show the details of the calculation of several evaluation metrics which are proposed in Guo et al. (2022a). The feature vectors of the ground-truth motion, the generated motion, and the text description are denoted by $f_{gt}$, $f_{gen}$ and $f_{text}$, respectively. Note that these features are extracted with the retrained evaluator from Guo et al. (2022a) using **MotionWeb** dataset.

#### A.3.1 FRECHET INCEPTION DISTANCE (FID)

Frechet Inception Distance quantifies the difference between the feature distributions of generated and real motion sequences. It is computed as follows:

$$\text{FID} = \|\boldsymbol{\mu}_{\text{gen}} - \boldsymbol{\mu}_{\text{gt}}\|^2 - \text{Tr}\left(\boldsymbol{\Sigma}_{\text{gen}} + \boldsymbol{\Sigma}_{\text{gt}} - 2\left(\boldsymbol{\Sigma}_{\text{gen}}\boldsymbol{\Sigma}_{\text{gt}}\right)^{\frac{1}{2}}\right), \tag{13}$$

where $\boldsymbol{\mu}_{\text{gen}}$ and $\boldsymbol{\mu}_{\text{gt}}$ represent the mean of the generated motion $f_{gen}$ and ground-truth motion $f_{gt}$, respectively. $\boldsymbol{\Sigma}_{\text{gen}}$ and $\boldsymbol{\Sigma}_{\text{gt}}$ denote the corresponding covariance matrices, while $\text{Tr}$ stands for the trace of a matrix.

#### A.3.2 R-PRECISION (TOP-1, TOP-2, TOP-3)

The R-Precision metric evaluates the alignment between motion and text by measuring the retrieval accuracy of ground-truth descriptions. Specifically, for each generated motion sequence, a candidate pool is constructed, consisting of its ground-truth text description and 31 randomly sampled mismatched descriptions from the dataset. The evaluation process involves the following steps:

1. Compute the Euclidean distances between the feature vector of the generated motion $f_{gen}$ and the feature vectors of the text descriptions $f_{text}$ for all candidates in the pool.

2. Rank the candidate descriptions in ascending order based on their Euclidean distances.

3. Calculate the average retrieval success rate across the Top-1, Top-2, and Top-3 rankings. A successful retrieval is defined as the ground-truth description appearing within the top $k$ candidates.

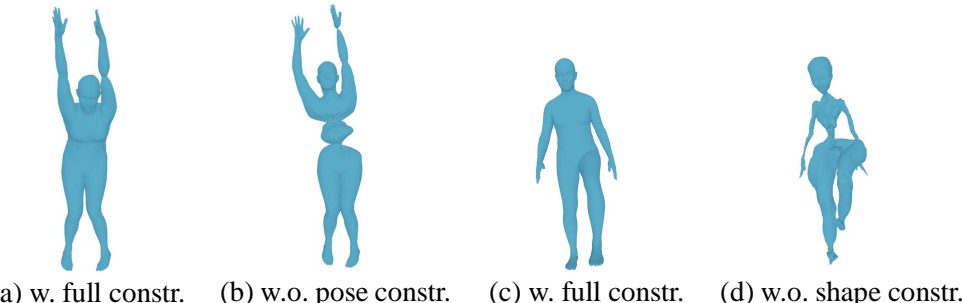

(a) w. full constr.    (b) w.o. pose constr.    (c) w. full constr.    (d) w.o. shape constr.

Figure 9: **Visualizations without constraint.** (a) and (c) are the control groups. (b) is the result without pose constraint and (d) is the result without shape constraint.

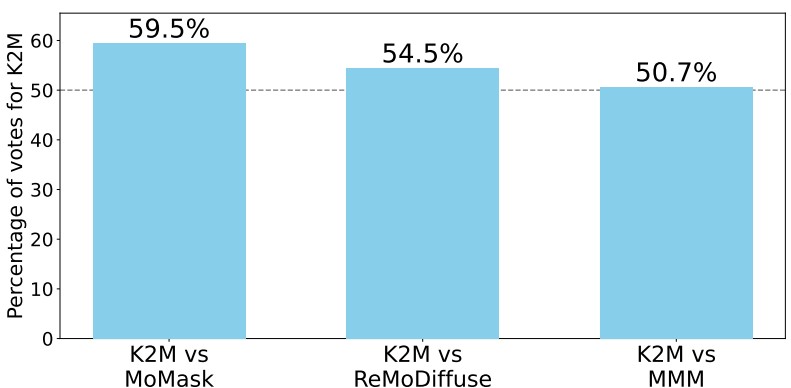

Figure 10: **User study.** Each bar represents the percentage of votes for **K2M** over the compared model. The dashed line marks 50%.

### A.3.3    MULTIMODAL-DISTANCE (MM-DIST)

This metric measures the distance between the generated motion feature and its corresponding text feature. It is calculated as:

$$\text{MM-Dist} = \|f_{gen} - f_{text}\|^2. \tag{14}$$

A lower MM-Dist value indicates a higher degree of alignment between motion and text.

### A.3.4    DIVERSITY

Diversity measures the variation in generated motion sequences across the dataset. It is computed by randomly selecting $N$ pairs of motion, where the $i$-th pair is denoted as $(f_{\text{gen}}^{i_1}, f_{\text{gen}}^{i_2})$. The average pairwise feature distance is then calculated as follows:

$$\text{Diversity} = \frac{1}{N}\sum_{i=1}^{N} \left\| f_{\text{gen}}^{i_1} - f_{\text{gen}}^{i_2} \right\|, \tag{15}$$

where $N$ is set to 300.

### A.4    EXPERIMENT DETAILS

Here, we provide additional experimental details not mentioned in the main text:

- The experiments were conducted on an NVIDIA A40 GPU with an Intel Xeon Gold 6330 CPU. Training the VQ-VAE model required approximately 36 hours, with a memory consumption of around 30 GB under a batch size of 256. For the transformer components,

the masked transformer and residual transformer each took roughly 6 hours to train, with a memory usage of approximately 36 GB when using a batch size of 64. All experiments were conducted 10 or more times, and the average values were reported as the final results.

- The hyperparameters $\lambda_{\text{rvq}}, \lambda_{\text{rec}}, \lambda_{\text{smooth}}, \lambda_{\text{vel}}, \lambda_{\text{boundary}}, \lambda_{\text{pose}}, \lambda_{\text{shape}}$ are set to 0.02, 0.02, 0.0001, 0.02, 0.0005, 10, and 1 respectively. This configuration was determined based on the actual numerical values of the loss terms. Our objective is to ensure that, after multiplying by the hyperparameters, the RVQ loss and reconstruction loss constitute the dominant components of the total loss, while the other loss terms remain at comparable magnitudes during the initial training phase.

- For visualization part, we use an RTX 3090 GPU and an AMD EPYC 7763 64-Core Processor.

- When computing motion sequence scores using MotionCritic Wang et al. (2024a), each sequence is truncated to the first 60 frames, as MotionCritic performs best when evaluating 60-frame sequences. All methods were fairly evaluated in the same setting.

- The frame rates (fps) of the visualizations provided by the AMASS dataset Mahmood et al. (2019) do not always match those of the motion sequences. For example, some visualization videos have only 5 fps, while the motion sequences are uniformly sampled at 20 fps. Additionally, the HumanAct12 dataset Guo et al. (2020) does not provide visualization data, further reducing the size of our training data. Despite this discrepancy, our model, trained on these visualizations, still generates motion sequences with the highest scores.

## A.5 USE OF LARGE LANGUAGE MODELS (LLMS)

In the preparation of this manuscript, we utilized Large Language Models (LLMs) as a writing assistant tool. The application of these models was strictly limited to enhancing the quality of the written English. Specifically, we employed them to aid in polishing the text for greater clarity, improving sentence structure, and ensuring a consistent linguistic flow throughout the document.

