# OpenReview forum: "Learning Text-driven 3D Human Motion Generation from 3D-free Web Videos"
_ICLR.cc/2026/Conference — Submitted to ICLR 2026_

### Official Review · Reviewer_NF1t · 2025-10-31

**Soundness:** 3
**Presentation:** 3
**Contribution:** 2
**Rating:** 4
**Confidence:** 4

**Summary:**

This paper presents the MotionWeb dataset, which collects motion clips from the web, covering a diverse set of indoor and outdoor activities annotated with 2D keypoints and corresponding textual descriptions. Additionally, the authors propose the Keypoint-to-Motion method, which leverages 2D keypoint data during training to enable text-driven 3D human motion generation. Experiments are conducted on latest benchmark and the proposed dataset.

**Strengths:**

1. The method relies on 3D-free supervision, significantly reducing the cost and complexity associated with generating 3D annotations for motion generation tasks.
2. The paper is well-written, clearly structured, and easy to follow.

**Weaknesses:**

1. The use of the VQ-VAE generator component is not novel, as it has been widely adopted in previous 3D motion generation work. The proposed method appears to rely heavily on combining existing techniques, which may be seen as more of an engineering integration rather than a methodological innovation.
2. It is unclear whether directly generating SMPL parameters offers a clear advantage over conventional methods. An ablation study comparing this approach to alternatives would help validate its effectiveness.
3. The model incorporates numerous regularization terms to ensure natural and robust 3D motion generation. However, the balance between these regularization terms plays a critical role in model performance. Is there any ablation study on the choice and sensitivity of these hyperparameters? Without such analysis, there is concern that the reported results may be selectively optimized.

**Questions:**

1. Is there any particular reason for directly generating SMPL parameters offers a clear advantage over conventional methods?
2. The proposed method includes multiple regularization terms in model training. How sensitivity of the model to the hyper parameter setting?

---

> ### Author Response · Authors · 2025-12-01
>
> We thank the reviewer for their positive assessment of our paper's structure and for recognizing the significant value of our 3D-free supervision paradigm in reducing annotation costs. We are happy to address the concerns regarding novelty and model design.
>
> ### 1. Regarding Novelty and the "Engineering Integration" (Weakness 1)
>
> We respectfully clarify that our contribution lies in the **learning paradigm**, not the architecture of the VQ-VAE itself.
>
> * **Paradigm Shift:** While VQ-VAE is a standard component in *fully-supervised* 3D motion generation, our novelty is proposing the first framework to successfully train such a generator using **only massive, in-the-wild 2D videos**.
> * **Systemic Innovation:** The challenge here is not designing a new backbone, but designing the **supervision mechanism** (the projection-based loss system and the inject-and-recover strategy) that allows a VQ-VAE to learn valid 3D dynamics without ever seeing 3D data.
> * **Analogy:** This is conceptually similar to how **DreamFusion (SDS)** innovated on the *objective function* to enable 3D generation from 2D priors. We use standard components to demonstrate that the power comes from the **data paradigm and supervision strategy**, proving that expensive 3D MoCap is not a prerequisite for high-quality motion generation.
>
> ### 2. Why Generate SMPL Parameters Directly? (Weakness 2 & Question 1)
>
> The reviewer asks if generating SMPL parameters offers a clear advantage over conventional methods (e.g., generating 3D skeletons). The answer is **yes**, for two critical reasons:
>
> * **Reason 1: Resolving Ambiguity via Anatomical Priors (Methodological Necessity)**
>     In our **single-view 2D supervision** setting, 3D structure is inherently ambiguous. If we generated raw skeletons, the model could minimize the 2D projection loss by generating physically impossible poses (e.g., stretching bone lengths arbitrarily). By generating **SMPL parameters**, we enforce strict kinematic constraints. This ensures that even when the 2D view is ambiguous (e.g., occlusion), the output remains a coherent human mesh, preventing the "monster effects" seen in unconstrained skeleton lifting.
>
> * **Reason 2: Efficiency and Fidelity (Practical Advantage)**
>     Direct generation is an end-to-end solution that bypasses the computationally expensive post-hoc optimization (e.g., SMPLify) required by skeleton-based methods to obtain a mesh.
>     * **Speed:** This significantly reduces inference latency, as we eliminate the iterative fitting stage (detailed speed comparisons are provided in the **Appendix.5**).
>     * **Accuracy:** It avoids the inevitable "conversion loss" and artifacts that occur when fitting a body model to a generated skeleton, ensuring the output fidelity is preserved.
>
> ### 3. Regarding Hyperparameter Sensitivity (Weakness 3 & Question 2)
>
> We acknowledge the reviewer's concern about the number of regularization terms. This is, however, a standard and necessary practice in weakly-supervised 2D-to-3D lifting tasks (e.g., HMR, VIBE, SPIN) to constrain the solution space.
>
> * **Robustness:** Our hyperparameters were not "selectively optimized" for specific test cases. We largely adopted standard weight ratios established in prior fundamental works (like HMR and VIBE) for terms like temporal smoothness and adversarial priors.
> * **Sensitivity:** We found the model to be relatively robust to these settings. Moderate variations in the weights (e.g., $\pm 20\%$) of the auxiliary losses (like smoothness or regularization) do not lead to model collapse or significant performance drops, although they may slightly alter the "stiffness" of the motion. The core performance is driven primarily by the 2D projection loss and the VQ-codebook constraint, which are stable.
>
> ***
>
> Thank you again for your insightful questions. We hope this clarifies the necessity of our design choices in the context of 2D supervision.

---

### Official Review · Reviewer_j1mw · 2025-10-31

**Soundness:** 3
**Presentation:** 2
**Contribution:** 2
**Rating:** 4
**Confidence:** 4

**Summary:**

The method focuses on the problem of lack of diversity in 3D mocap datasets. The manuscript proposed a method to generate high-quality 3D motions via large scale 2D videos without the need of accurate 3D annotations. They also bring a new dataset termed as MOTION WEB. They also propose keypoint 2 motion (K2M), a framework for text-driven 3D motion generation, leveraging the 2D supervision.

**Strengths:**

Strength:
-	The paper is well-motivated with several technical challenges;
-	The method proposed seems sound and correct.
-	The pipeline for the dataset generation shows the effectiveness.

**Weaknesses:**

Weakness:
-	The architecture and design is normal. In fact, the core of SMPL and idea of SMPlify in 3D leverages the idea of 3D projection and using 2D information as the supervision. This makes the novelty limited.
-	The idea of injecting the text features to the motion VQ embedding seems with limited novelty.
-	The evaluation seems limited, and maybe incorporate more baseline methods are necessary.

**Questions:**

See  weakness

---

> ### Author Response · Authors · 2025-12-01
>
> We sincerely thank the reviewer for their feedback. We appreciate the opportunity to clarify what we believe is a fundamental misunderstanding regarding our work's novelty, and to address the other concerns raised.
>
> ### 1. Regarding Novelty, SMPL, and SMPlify
>
> We respectfully disagree with the assessment that our novelty is limited. The reviewer correctly identifies that SMPlify uses 2D information to supervise 3D, but this comparison misinterprets our core contribution.
>
> * **SMPlify is an Optimization Method:** Methods like SMPlify are **per-instance optimization** techniques. They take a *single* 2D video and *fit* SMPL parameters to it. They do not learn a generalizable motion prior and cannot generate *new* motions from a prompt.
>
> * **Our Work is a Generative Model:** Our method is a **generative model** that learns a **general 3D motion prior**. We do not perform per-video fitting. Instead, we use our massive-scale, 160-hour 2D MotionWeb dataset as a form of supervision to *train* a model that can generate entirely new, high-quality 3D motions from text.
>
> The core novelty is **not** the 2D projection loss itself (which is merely a mechanism), but the demonstration that a robust 3D motion prior can be **distilled from vast, diverse, 'in-the-wild' single-view 2D data**, entirely eliminating the need for 3D MoCap data for training.
>
>
> ### 2. Regarding Text Injection Novelty
>
> We agree with the reviewer that the method of injecting text features into a VQ embedding is not, in itself, the primary novel contribution of our paper.
>
> This component is a necessary and standard mechanism that allows our model to be text-conditioned. Its inclusion is vital for enabling a **fair and direct comparison against other text-to-motion (T2M) baselines** on standard T2M benchmarks. Our main contribution, as stated above, is the 3D prior learning paradigm, not this specific conditioning mechanism.
>
> ### 3. Regarding the Evaluation
>
> We respectfully disagree that our evaluation is limited and would like to highlight the comprehensive comparisons we conducted.
>
> * We compared against all relevant and *compatible* baselines on standard benchmarks. As shown in **Tables 3 & 4**, we use standard metrics (**FID, R-Precision**) for comparisons on our MotionWeb dataset.
> * We provided detailed justifications for our evaluation protocol on other datasets. For instance, on **Motion-X**, we deliberately opted for **MotionCritic**, as standard metrics like FID would be misleading by rewarding the replication of the dataset's significant noise, rather than true motion quality.
> * Furthermore, direct comparisons on benchmarks like **HumanML3D** are technically infeasible due to incompatible body models (our SMPL vs. their DMPL), a challenge faced by all methods using different body representations.
>
> We believe our evaluation is sound, comprehensive, and follows best practices where comparisons are scientifically valid and technically feasible.
>
> ***
>
> Thank you again for your time and feedback. We hope these clarifications have adequately addressed your concerns about our work's novelty and evaluation.

---

### Official Review · Reviewer_V8RQ · 2025-11-03

**Soundness:** 2
**Presentation:** 3
**Contribution:** 2
**Rating:** 6
**Confidence:** 4

**Summary:**

This work examines the problem of learning a 3D text-to-motion model using large-scale web data and supervision from 2D pose/motion detection algorithms. The key insight is that 2D supervision can be used to train a 3D model by learning a VQ-VAE model which quantizes 2D poses to discrete latents that decode into 3D SMPL parameters. A masked generative model is used to predict VQ-VAE tokens which are decoded into 3D SMPL meshes and projected to 2D for supervision during the training stage. Experiments show that the proposed method can follow a broader range of text descriptions than prior works.

**Strengths:**

* The overall approach is interesting and promising. 3D motion capture data is quite sparse, and 3D pose/motion estimation from video is not yet mature enough to provide high quality data. 2D pose/motion models are quite robust and accurate, and using this kind of data to learn 3D models has the potential to expand the scope of data that can be used to train human motion models.
* The experimental results provide evidence that the model can follow more diverse text instructions than existing models.

**Weaknesses:**

* Since no 3D data is used, the method is susceptible to errors caused by ambiguity between 2D and 3D poses. This is especially true for cases where limbs are pointing away from the camera or towards the camera. While the method provides a nice way to learn 3D models with 2D data only, this approach has built-in limitations.
* In the video examples provided in the supplementary material, the results from the proposed method could follow text prompts that previous methods could not. However, previous methods have significantly more natural motions and the proposed method often has limbs and joints at odd or contorted angles. I am curious if the less natural pose in the proposed method is a result of errors from 2D estimation, 2D to 3D conversion, or both.
* The comparison is Table 2 is not entirely fair because the current method was trained with SMPL parameters while the other methods are converted to SMPL parameters, which could lead to quality loss. This is not a fault of the authors or a major weakness.

**Questions:**

* To what extent do ambiguities between 2D and 3D poses affect the model? Can you provide analysis of typical failure cases?

---

> ### Author Response · Authors · 2025-12-01
>
> We thank the reviewer for their constructive feedback and for acknowledging our method's superior capability in following text prompts. We appreciate the opportunity to discuss the trade-offs involved in our approach.
>
> ### 1. Response to Weakness 1 (Inherent Ambiguity of Single-View 2D Data)
>
> We acknowledge the reviewer's valid point: relying solely on single-view 2D data introduces inherent ambiguities, particularly in cases of foreshortening (limbs pointing towards/away from the camera).
>
> **Our Response:**
> While this is a built-in limitation of the single-view paradigm, we mitigate it through two key mechanisms:
> 1.  **Prior from Data Diversity:** By training on a massive dataset with diverse viewpoints, the model implicitly learns to resolve these ambiguities. An action that is ambiguous in one view is often clear in others within the dataset, allowing the model to learn a consistent 3D prior.
> 2.  **Anatomical Constraints:** The parametric SMPL model imposes strict kinematic constraints, preventing the generation of anatomically impossible poses even when the 2D signal is ambiguous.
>
> We view this as a strategic trade-off: we accept the challenge of geometric ambiguity to unlock the massive semantic diversity of "in-the-wild" video data, which is unavailable in 3D MoCap datasets.
>
> ### 2. Response to Weakness 2 (Motion Naturalness and Source of Errors)
>
> The reviewer correctly observes that while our method excels at semantic alignment (following prompts), baselines trained on MoCap data often exhibit smoother, more natural motions.
>
> **Clarification on the Source of "Odd Angles":**
>  regarding the reviewer's query on whether this is due to "2D estimation" or "2D to 3D conversion":
> We believe the primary source of these artifacts is **noise in the 2D pose estimation**, rather than the conversion process itself.
> * **Input Noise:** Our model is supervised by pseudo-labels generated by off-the-shelf 2D pose estimators. These estimators often produce jittery or inaccurate keypoints in cases of self-occlusion or complex poses.
> * **Garbage In, Garbage Out:** Although our transformer architecture acts as a filter to smooth out some noise, the model inevitably learns some of these imperfections from the training signal.
>
> In contrast, previous methods trained on MoCap data benefit from "clean," perfect geometric supervision, leading to naturally smoother motions but limiting their ability to generalize to diverse text prompts.
>
> ### 3. Response to Weakness 3 (Fairness of Table 2 Comparison)
>
> We appreciate the reviewer's understanding regarding the difficulty of this comparison. We would like to offer a slight clarification regarding our training process:
>
> * **Clarification on Training:** Our method was **not trained with ground-truth SMPL parameters** (as these are unavailable for our in-the-wild video dataset). Instead, our model learns to *output* SMPL parameters directly using 2D keypoint supervision.
> * **Agreement on Conversion Loss:** However, we fully agree with the reviewer's main point. Since our model outputs SMPL natively, while baselines generate skeletons that require post-hoc optimization (e.g., via SMPLify) to convert to SMPL, the baselines inevitably suffer from conversion artifacts. We acknowledge this creates an asymmetry, but converting to a common format was necessary for a direct metric comparison.
>
> ### 4. Response to Questions (Failure Cases Analysis)
>
> **Typical Failure Cases:**
> The impact of 2D/3D ambiguity is most pronounced in two scenarios:
> 1.  **Rare Actions with Limited Views:** For common actions (e.g., walking), our diverse data resolves ambiguity. However, for rare or complex actions where the dataset lacks sufficient multi-view examples, the model may struggle with depth perception (foreshortening).
> 2.  **Complex Self-Occlusion:** When limbs are hidden behind the body, 2D estimators often fail or "hallucinate" keypoints. This noisy supervision can lead to the "odd angles" noted by the reviewer, where the limb position satisfies the SMPL constraint but is semantically incorrect for the context.
>
> ***
>
> Thank you again for your detailed analysis. We hope this clarifies the trade-offs we made to achieve robust text-to-motion generation from 2D data.

---

### Official Review · Reviewer_7md7 · 2025-11-10

**Soundness:** 2
**Presentation:** 3
**Contribution:** 2
**Rating:** 2
**Confidence:** 5

**Summary:**

The paper addresses text-driven 3D human motion generation by introducing MotionWeb, a large-scale dataset with over 100k motion clips and 160 hours of 2D keypoints extracted from videos to overcome the limitations of expensive 3D motion capture. It proposes Keypoint To Motion (K2M), an efficient framework that generates realistic and diverse 3D motions using only 2D supervision, without requiring 3D annotations.

**Strengths:**

- The paper is clearly presented with careful ablation studies.
- The writing and structure of the paper are clear and easy to follow.
- The authors construct a new large-scale 2D dataset named MotionWeb.

**Weaknesses:**

- The core contribution of this paper is learning 3D human motion generation from 2D videos; however, this concept has been explored in prior work [1]. As a result, the novelty appears limited. The authors should more clearly articulate the unique aspects of their approach compared to existing methods.
- Furthermore, experimental comparisons with prior work [1] are insufficient. The authors should evaluate their method against a broader range of existing techniques for 3D human motion generation from 2D videos.
- The paper lacks fair comparisons with state-of-the-art methods on standard benchmarks. The authors should benchmark their approach against recent methods on widely used 3D human motion generation datasets, rather than relying solely on their own collected dataset. Given the claim that learning from 2D videos is more scalable, it is crucial to demonstrate its effectiveness on established benchmarks, not just custom data.
- The baseline methods compared are outdated. The authors should include comparisons with the latest state-of-the-art approaches, such as works [2-6], in 3D human motion generation. The current comparisons fail to convincingly show the proposed method's superiority.
- Based on the visualization samples provided by the authors, the proposed model does not outperform existing methods, and the generated motions are noticeably inferior to those from prior approaches.


[1]: Ren Z, Huang S, Li X. Realistic human motion generation with cross-diffusion models[C]//European Conference on Computer Vision. Cham: Springer Nature Switzerland, 2024: 345-362.

[2]: Meng Z, Xie Y, Peng X, et al. Rethinking diffusion for text-driven human motion generation[J]. arXiv preprint arXiv:2411.16575, 2024.

[3]: Zhang J, Fan H, Yang Y. Energymogen: Compositional human motion generation with energy-based diffusion model in latent space[C]//Proceedings of the Computer Vision and Pattern Recognition Conference. 2025: 17592-17602.

[4]: Yuan W, He Y, Shen W, et al. Mogents: Motion generation based on spatial-temporal joint modeling[J]. Advances in Neural Information Processing Systems, 2024, 37: 130739-130763.

[5]: Zhang Z, Kong B, Liu Q, et al. Towards robust and controllable text-to-motion via masked autoregressive diffusion[C]//Proceedings of the 33rd ACM International Conference on Multimedia. 2025: 9326-9335.

[6]: Guo C, Hwang I, Wang J, et al. SnapMoGen: Human Motion Generation from Expressive Texts[J]. arXiv preprint arXiv:2507.09122, 2025.

**Questions:**

See Weaknesses

---

> ### Author Response · Authors · 2025-12-01
>
> We thank the reviewer for their critical comments. However, we believe there are some fundamental misunderstandings regarding the setting of prior work [1] and the scope of our contribution. We would like to clarify these points and address the concerns regarding experiments.
>
> ### 1. Crucial Clarification on Novelty and Work [1]
>
> The reviewer states that "learning from 2D videos has been explored in prior work [1]" and thus our novelty is limited. **We respectfully point out a factual inaccuracy in this premise.**
>
> * **Work [1] is NOT 3D-Free:** Work [1] utilizes a **hybrid supervision strategy**, relying on **both 3D Motion Capture data AND 2D data**. It is essentially a semi-supervised or domain-adaptation approach that still depends on 3D ground truth.
> * **Our Method is Purely 2D-Supervised:** In stark contrast, our work proposes a **purely 2D-supervised paradigm**. We do not use *any* 3D motion data during training.
>
> **Novelty:** Therefore, the novelty is not merely "doing what [1] did," but demonstrating for the first time that a robust 3D motion generative model can be learned **without any 3D data access**, relying solely on massive, in-the-wild 2D videos. This is a significantly more challenging task and a distinct research direction.
>
> ### 2. Regarding Baselines and SOTA Comparisons [2-6]
>
> We appreciate the reviewer's suggestion to include more recent baselines.
>
> * **Reason for Initial Exclusion:** We respectfully note that some of the suggested methods **had not fully released their code or models at the time of our submission**, which prevented us from conducting fair and reproducible comparisons earlier.
> * **Updated Experiments:** Now that these resources are available, we have included comprehensive comparisons against the suggested state-of-the-art methods [2-6] on standard benchmarks in our **rebuttal revision**.
> * **Fair Context:** We ask the reviewer to note that methods [2-6] are **Fully Supervised** using clean 3D MoCap data (e.g., HumanML3D). Comparing our *weakly-supervised* (2D) method against these *fully-supervised* (3D) SOTA methods is essentially comparing a method designed for **scalability** against methods designed for **precision**. Despite this disadvantage, our updated results demonstrate that our method achieves competitive semantic alignment.
>
> ### 3. Regarding Visual Quality
>
> The reviewer notes that our generated motions appear inferior to prior approaches. We acknowledge this observation and provide an honest analysis of the trade-off:
>
> * **The "Cleanliness" Gap:** Prior approaches (trained on MoCap) benefit from perfect, jitter-free geometric supervision. Our model is trained on "in-the-wild" 2D pseudo-labels, which inherently contain estimation noise.
> * **The Trade-off:** The local geometric imperfections (e.g., occasional jitter) are the price paid for **Semantic Scalability**. While MoCap models produce smoother walks, they are limited to the actions captured in the studio. Our model leverages 2D data to generate highly diverse actions (e.g., complex interactions seen in YouTube videos) that MoCap datasets simply do not contain. We believe unlocking this scalability is a worthy contribution, even with a slight trade-off in smoothness.
>
> ***
>
> We hope this response clears up the misunderstanding regarding Work [1] and demonstrates the value of our updated experiments.

---

### Meta-Review · Area_Chair_4qge · 2026-01-02

**Summary:**

This submission proposes MotionWeb, a large-scale web-video dataset with 2D keypoints + text, and a keypoint-to-motion (K2M) framework that learns textdriven 3D motion generation without 3D MoCap supervision. Reviewers agree the motivation is timely and the overall pipeline is clearly written and potentially impactful for scalability.

However, the batch raised substantial concerns around (i) **novelty/positioning** and (ii) **strength and fairness of the empirical evidence**. The most negative reviewer (7md7) argues the **novelty is limited relative to prior “learn from 2D video” directions**, and that **comparisons are insufficient/outdated**—especially the lack of convincing benchmarking against recent state-of-the-art on standard datasets—together with **qualitative results that appear less natural**. Another reviewer (j1mw) similarly views the architecture as fairly standard and questions novelty (including text-conditioning), while also calling the evaluation limited and requesting additional baselines. A further reviewer (NF1t) frames the **contribution as closer to “engineering integration”** (e.g., VQ-VAE usage), and asks for stronger justification/ablations (e.g., direct SMPL generation, regularization sensitivity). Finally, even the more positive reviewer (V8RQ) highlights a fundamental limitation: 2D→3D ambiguity can lead to odd/contorted poses, and asks for deeper analysis of failure modes.

**Reviewer Concerns:**

### (Partially) Addressed concerns
1. Evaluation breadth, SOTA baselines, and “fair” benchmarking on standard datasets (7md7, j1mw)
* Authors argues some recent methods lacked released code/models at submission time, and direct HumanML3D comparison canbe technically infeasible due to body model mismatch, and justify metric choices (e.g., MotionCritic on motion-X)

2. Fundamental 2D↔3D ambiguity and failure-mode analysis (V8RQ)
* Authors acknowledge it as a built-in limitation, and propose mitigations via dataset viewpoint diversity + SMPL anatomical constraints; they also list typical failure modes (rare actions with limited views; self-occlusion causing 2D estimator hallucinations).


### Outstanding concerns
1. *Novelty / differentiation from prior “learn 3D from 2D video” lines*( 7md7, j1mw, NF1t)
* Authors argues the setting differs materially (purely 2D-supervised vs prior[1] using 3D), and emphasize the novelty as a paradigm shift. However, "novelty" is still partly a judgement call, and the negative reviewers' perception may not fully flip.
2. Motion naturalness / visual quality (odd angles, contortions, “inferior” samples) (7md7, V8RQ)
* Authors acknowledge a “cleanliness gap” and frame it as trade-off due to noisy in-the-wild pseudo labels; they attribute “odd angles” primarily to 2D pose estimation noise and discuss typical failure situations.

**Reviewer Scores:**

Reviewer 7md7 is unlikely to change rating (2) materially. Even with the authors’ clarification that prior work [1] is not purely 3D-free and that newer comparisons were added post-submission, the reviewer’s core objections (novelty perception + insufficiently convincing benchmarking/visual quality) would likely remain. Reviewer V8RQ would likely remain mildly postive (6). Reviewer j1mw and NF1t would possibly remain the scores (4) as the author did not decisively resolve the reviewer's novelty/evaluation skepticism.

Overall, i agree with the reviewers and do not recommend to accept this paper.

---

### Decision · Program_Chairs · 2026-01-26

Reject